# A meta-analysis of plant facilitation in coastal dune systems: responses, regions, and research gaps

Camila de Toledo Castanho[1,*], Christopher J. Lortie[2], Benjamin Zaitchik[3] and Paulo Inácio Prado[1]

[1] Departamento de Ecologia, Instituto de Biociências, Universidade de São Paulo, São Paulo, Brazil
[2] Department of Biology, York University, Toronto, Ontario, Canada
[3] Department of Earth and Planetary Sciences, Johns Hopkins University, Baltimore, MD, USA
[*] Current affiliation: Departamento de Ciências Biológicas, Universidade Federal de São Paulo, Diadema, São Paulo, Brazil

## ABSTRACT

Empirical studies in salt marshes, arid, and alpine systems support the hypothesis that facilitation between plants is an important ecological process in severe or 'stressful' environments. Coastal dunes are both abiotically stressful and frequently disturbed systems. Facilitation has been documented, but the evidence to date has not been synthesized. We did a systematic review with meta-analysis to highlight general research gaps in the study of plant interactions in coastal dunes and examine if regional and local factors influence the magnitude of facilitation in these systems. The 32 studies included in the systematic review were done in coastal dunes located in 13 countries around the world but the majority was in the temperate zone (63%). Most of the studies adopt only an observational approach to make inferences about facilitative interactions, whereas only 28% of the studies used both observational and experimental approaches. Among the factors we tested, only geographic region mediates the occurrence of facilitation more broadly in coastal dune systems. The presence of a neighbor positively influenced growth and survival in the tropics, whereas in temperate and subartic regions the effect was neutral for both response variables. We found no evidence that climatic and local factors, such as life-form and life stage of interacting plants, affect the magnitude of facilitation in coastal dunes. Overall, conclusions about plant facilitation in coastal dunes depend on the response variable measured and, more broadly, on the geographic region examined. However, the high variability and the limited number of studies, especially in tropical region, indicate we need to be cautious in the generalization of the conclusions. Anyway, coastal dunes provide an important means to explore topical issues in facilitation research including context dependency, local versus regional drivers of community structure, and the importance of gradients in shaping the outcome of net interactions.

Corresponding author
Camila de Toledo Castanho,
ctcastanho@gmail.com

## INTRODUCTION

The role of positive interactions, or facilitation, between plants as key drivers of plant community dynamics and structure is widely recognized and reviewed (*Brooker et al., 2008*; *McIntire & Fajardo, 2014*). Most empirical studies show that facilitative effects are more important in severe environments because neighbors frequently buffer other individuals from abiotic stressors (*He, Bertness & Altieri, 2013*). The classical systems that generated this research topic are deserts (*Franco & Nobel, 1989*; *Flores & Jurado, 2003*), salt marshes (*Bertness & Hacker, 1994*; *Bertness & Leonard, 1997*), and more recently, alpine systems (*Badano et al., 2006*; *Cavieres et al., 2014*). However, positive interactions may also be important in many other ecosystems, and there are similar gradients that likely shift the relative frequency of positive interactions.

Recent research in coastal dune vegetation has increasingly focused on facilitation between plants. Coastal dune vegetation, here defined as a mosaic of plant communities in the coast that occupy sandy plains formed by marine deposits (modified from *Scarano, 2002*), is both stressful and highly disturbed, with soil moisture and nutrient limitations, wind exposure, sand burial, salt spray and soil salinity, potentially negatively impacting plants (*Wilson & Sykes, 1999*). Similar to desert systems, the presence of some plants, such as shrubs, ameliorate some of these limiting factors and can provide an opportunity for association by other species (*Martínez & García-Franco, 2004*). Several studies in coastal dunes have shown that the performance of plants established in the neighborhood of other plants are higher than in open areas (*Shumway, 2000*; *Martínez, 2003*; *Forey, Lortie & Michalet, 2009*; *Castanho & Prado, 2014*). However, the occurrence and intensity of facilitation in coastal dunes is also highly variable within and between the studies (*Forey, Lortie & Michalet, 2009*; *Castanho, Oliveira & Prado, 2012*) thereby suggesting that facilitation is dependent on the local environmental conditions or the gradients (*He, Bertness & Altieri, 2013*) and also on the traits of interacting plants (*Soliveres et al., 2014*). As demonstrated in dunes and other systems, the magnitude of facilitation depends on plant life-stage (*Miriti, 2006*; *Armas & Pugnaire, 2009*) and plant life-form (*Gómez-Aparicio, 2009*; *Castanho, Oliveira & Prado, 2012*) with higher intensities associated with adult woody benefactors and woody beneficiary species at relatively earlier life-stages. Furthermore, the enviromental severity also shapes the outcome of interactions, with more intense facilitation commonly detected under increasingly harsh conditions (*He, Bertness & Altieri, 2013*). Consequently, coastal dunes may also be an ideal system to explore net interactions in plants communities. However, without synthesis, the context dependency of these positive interactions is not broadly accessible (*Gómez-Aparicio, 2009*) and research gaps are not easily identified.

Therefore, a formal quantitative analysis of the literature in these systems is required. Such analysis can provide an estimate of the general influence of facilitation on the organization and dynamics of the coastal dunes, further the scope of hypothesis testing in this ecological subdiscipline, and contrast the relative importance of local versus regional drivers of plant community structure (*Thebault et al., 2014*). We note that the

scale of drivers of community structure is an important contemporary issue in ecology (*Powers et al., 2009*; *O'Halloran et al., 2013*) that most likely needs to be resolved on an ecosystem-by-ecosystem basis. Moreover, the restoration of degraded coastal dunes is a pressing issue in many regions of the world (*Lithgow et al., 2013*), and facilitation by dominant coastal plant species is an obvious potential management solution.

To meet these research needs, we present a systematic review and meta-analysis of facilitation in coastal dune plant communities. The systematic review synthesizes current literature and highlights research gaps, while the meta-analysis tests if factors at distinct scales (local versus regional) such as environmental severity, life-form, or life-stage of the interacting plants significantly explained the variation in the intensity of plant facilitation in coastal dunes.

## MATERIALS & METHODS

### Data collection

We conducted a survey of the published studies that explicitly tested for the presence of facilitation between plants in coastal dunes. The literature was queried by using ISI Web of Science in June 2013 by using a combination of three groups of terms: (i) "dune*" or "restinga" or "coastal sand vegetation," and (ii) "facilitation" or "positive interaction*" and (iii) "plant*" or "tree*" or "shrub*" or "herb*." We did not include competition studies because our main aim was to test factors that affect the intensity of facilitation in dunes and not to make inferences about the importance of facilitation relative to competition. The search led to 90 publications that were subsequently examined firstly for their suitability in the review and secondly in the meta-analysis. For the first selection criterion, only studies that explicitly examined facilitation between plants in coastal dune vegetation under field conditions were included ($n = 32$). Reviews, studies on non-coastal dunes, and those in which at least one of the interacting organisms were not a plant were excluded. To conduct the meta-analysis, the studies also had to include the following: (i) data reported in a usable form; and (ii) the effect of neighbors on target species compared to the performance of plants without neighbors. When the required data were only reported in graphical form, the graphics were scanned and extracted in table format using TechDig software (*Jones, 1998*). Multiple outcomes per publication that tested different combinations of neighbor-target species, different life stages, or different sites were treated as independent outcomes. However, if repeated measures were taken from the same experiment, only the results reported at the completion of the experiment were used. Similarly, only the final year in multi-year experiments was used as a conservative estimate of impacts and to avoid pseudoreplication issues. Furthermore, if additional treatment such as water or fertility addition was performed, we only used the estimates from the control level (no addition) because it better approximates the natural/ambient conditions. Authors of publications with unreported datasets were also contacted to secure data.

To understand how facilitation intensity varies among study outcomes, each outcome was classified according to the following explanatory variables: neighbour and target life-forms, target life stage, geographic regions and environmental severity. Regarding

life-forms, neighbour and target plants from each outcome was classified as tree, shrubs or herbs (which was further subdivided into grass and forb when the information was available). For target life stage we classified the target plants as seed, young (including seedlings, saplings and juveniles) or adult. We also classified each outcome according to the geographic region, i.e., as tropical (from latitude 0° to 28°), temperate (29° to 54°) and subarctic-subantarctic (more than 55°) using the reported latitudes. Because of the coarse sand texture of the soils, water is often a limiting resource in coastal dunes (*Maun, 1994*; *Le Bagousse-Pinguet et al., 2013*). Therefore, we used mean annual precipitation (MAP) of each site as a proxy for environmental severity. Based on the GPS coordinates listed in each paper, we extracted an estimate of MAP for each study site from the meteorological forcing fields of the Global Land Data Assimilation System, version 1 (GLDASv1). GLDAS is a global, high-resolution terrestrial modelling system that incorporates satellite and ground-based observations in order to produce optimal fields of land surface states and fluxes in near–real time (*Rodell et al., 2004*).

While MAP is one indicator of environmental severity, plant life at coastal dunes around the world can be limited by a combination of factors such as nutrient limitation, salinity, and sand burial (*Maun, 1994*; *Wilson & Sykes, 1999*). The complex nature of coastal limiting factors typically justifies the use of integrative proxies for environmental severity such as plant biomass (*Dullinger et al., 2007*; *Maestre et al., 2009*). For this reason, we also used the normalized difference vegetation index (NDVI), a proxy for plant biomass (*Paruelo et al., 1997*; *Doiron et al., 2013*), as an integrative variable of environmental severity at both local and regional scales. To estimate the NDVI of each site, we used remote sensing techniques and two kinds of images with different resolutions: Advanced Spaceborn Thermal Emission and Reflection Radiometer (ASTER) images with 15 m resolution which provided a local estimate of biomass vegetation; and Moderate Imaging Spectroradiometer (MODIS) images with 250 m resolution which also provided a regional estimation. For the NDVI based on ASTER images (hereafter called local NDVI), we used the coordinates and description of the site (Reserve, National Park, etc.) and local vegetation provided by each study in order to place the study site within the image as precisely as possible. Then, the archive of ASTER images available for each site were searched and images were selected at the same time as the study implementation and also to minimize cloud cover. For the NDVI based on MODIS images (hereafter called regional NDVI), mensal images were used from 2001 to 2009. In order to synthesize this information for each site, the mensal NDVI was summed annually to calculate the mean annual NDVI. All NDVI calculation was done using the software ERDAS IMAGINE 2011 (Intergraph; Madison, AL, USA).

## Meta-analysis

Suitable studies were grouped into eight different datasets according to the plant response variable reported: density, survival, growth (which includes biomass and growth in height), richness (number of species), reproductive output (which includes any quantitative measure of flower, fruit or seeds production), occurrence, and emergence.

Because we had two types of response variables, we used different measures of effect size: the natural log of the response ratio (ln (RR)) for continuous response variables (density, growth, richness and reproductive output), and the natural log odds ratio (ln (OR)) for binomial response variables (survival, occurrence and seed emergence) (*Rosenberg, Rothstein & Gurevitch, 2013*).

The natural log of the response ratio (ln (RR)) estimation, and its associated variance, was calculated for each outcome using the mean, standard deviation (SD) and sample size (n) for control (without neighbor) and treatment (with neighbor) (*Rosenberg, Rothstein & Gurevitch, 2013*). Values of ln (RR) higher than 0 indicate a positive effect of the neighbor on the target performance (facilitation) whilst values lower than 0 indicate a negative effect of the neighbor (competition). For categorical responses, the natural log of the odds ratio (ln (OR)) and its associated variance for each outcome is calculated using the number of success and failure occurrences for each treatment (*Rosenberg, Rothstein & Gurevitch, 2013*). In the case of survival for example, this measure denotes the number of survival and dead plants with and without neighbors, positive values of ln (OR) also indicate facilitation. In the few cases where survival data were reported as mean and SD, we first calculated Hedges g, converted to Cohen d, and finally to the common index ln (OR) in order to combine all survival outcomes in the same meta-analysis (*Borenstein et al., 2009*).

The effect of the neighbor was assessed for each one of the response variable datasets that included in at least five independent studies (this was a conservative threshold to ensure general value to the synthesis). We used a threshold to avoid potential biases from trends associated to too few studies. Bias-corrected bootstrap 95% confidence intervals (CIs) were calculated for each overall effect size. If the CI did not overlap zero, the effect was considered significant (*Rosenberg, 2013*). The Q-statistics were used for each dataset in order to examine the heterogeneity among the effect sizes, and the proportion of true variance in the effect sizes explained by each independent variable was estimated as $R^2$ (*Borenstein et al., 2009*). The significance of the model structure was tested by randomization tests with 9,999 iterations ($\alpha = 0.05$). The independent variables selected were (i) geographic region (tropical, temperate or subarctic-subantarctic regions), (ii) neighbor life-form (i.e., tree, shrub, grass or forb); (iii) target life-form (i.e., tree, shrub, grass or forb); (iv) target life stage; (v) mean annual precipitation (MAP); (vi) local NDVI; and (vii) regional NDVI.

Data were analyzed using mixed-effect models that encompass both fixed and random effects, with fixed differences among predictors (continuous or categorical covariates) and random variation among studies within levels of the predictor, as well sampling error within studies (*Mengersen et al., 2013*). In the present context, the random variation among studies are more reasonable than fixed variation because the complex interactions in ecology generally result in ecologically important heterogeneity between studies (*Pullin & Stewart, 2006*). Additionally, the use of mixed-effect models (with random variation among studies instead of fixed variation) fits the goal of generalization usually present in similar reviews (*Borenstein et al., 2009*). We tested publication bias calculating the

Rosenthal's fail number, specifically, that a fail-safe number larger than $5n + 10$ (where $n$ is the number of outcomes) is a conservative critical value (*Rosenthal, 1979*). Funnel plots with Kendall's tau rank correlation tests were also examined to explore potential publication bias. We used the R environment (version 3.1, *R Core Team, 2014*) with the package metafor (*Viechtbauer, 2010*) for all statistical analysis.

## RESULTS

### Systematic review

From 90 articles identified through database searching, a total of 32 articles met the selection criteria for the systematic review (Fig. 1 and Table S1). These articles were published in 16 different journals between 1997 and 2013. These studies were performed in 13 countries, but a total of 31% of all studies were done on coastal dunes in the USA. With respect to diversity of climatic zones examined, 63% of the studies were done in the temperate zone, 28% in the tropics, and 9% in the arctic-subarctic zone. A total of 15 studies, i.e., 47%, were observational and 8 (25%) were manipulative whilst 9 studies (28%) used both approaches. A total of 362 independent outcomes were extracted for the seven plant performance response variables. Density was the most frequent representing 31% of the total, followed by survival (22%), growth (18%), reproductive output (12%), richness (8%), occurrence (7%) and emergence (2%). The earlier life stages for target plant species were most represented in these studies (i.e., seedlings, saplings, and juveniles) with 43% of the total number of measurements versus 20% of studies recording adults. Shrubs were the most common nurse-plants examined representing 46% of the outcomes, followed by herb (36%), tree (9%), moss and lichen (4%) and a mix of life-forms used in 4% of the outcomes. The most common target life-forms were herb (51%), followed by mix of life-forms (24%), shrub (19%), tree (4%), liana (1%) and moss (1%).

### Meta-analysis

A total of 160 independent effect size estimates were suitable for the meta-analysis component of this synthesis (Table 1). Density (Table S2), growth (Table S3) and survival (Table S4) datasets provided sufficient independent studies, i.e., at least 5 studies and 10 effect sizes (outcomes), to be considered in the meta-analysis (Table 1). There was no evidence of publication bias for density and growth (Kendall's tau $= 0.14$, $P = 0.22$ for density; Kendall's tau $= 0.06$, $P = 0.58$ for growth) and a limited indication of bias for survival (Kendall's tau $= -0.23$, $P = 0.04$). The fail-safe numbers indicated that the results detected for density and survival could be driven by limited or biased sets of publications available for synthesis (for density: critical threshold $= 210$, fail-safe number $= 181$; for growth: critical threshold $= 205$, fail-safe number $= 468$, for survival: critical threshold $= 220$, fail-safe number $= 40$).

Across all study outcomes, the presence of neighbor had no effect on the density of the target species examined (mean lnRR $= 0.38$, 95% confidence intervals: $-0.13$–$0.88$). However, the overall heterogeneity test was significant indicating that the different study outcomes do not share a common effect and explanatory variables may explain the

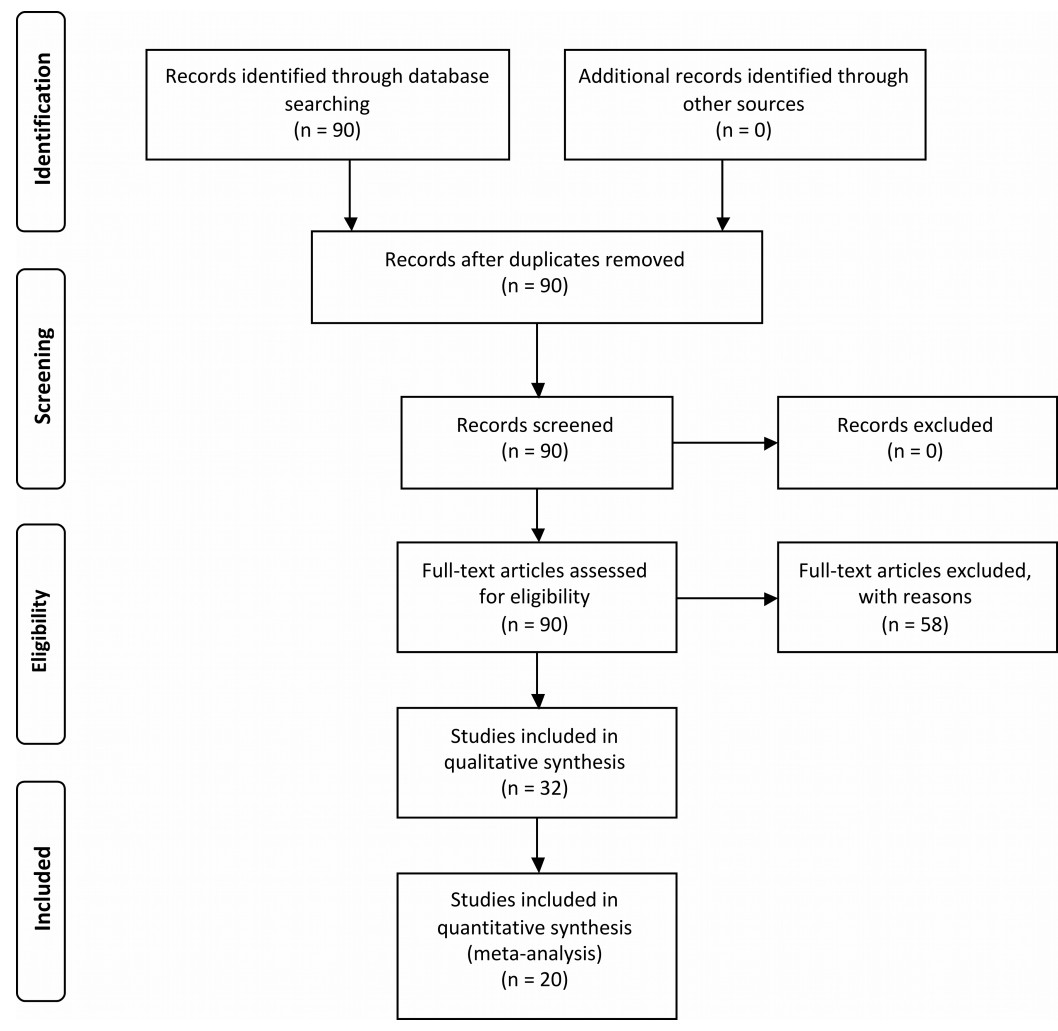

**Figure 1** Prisma flow diagram (*Moher et al., 2009*) depicting the seach protocol and workflow in determining the effective population of studies for systematic review and meta-analysis.

**Table 1** **Meta-analysis studies.** The total number of suitable independent cases for meta-analysis (outcomes) and the respective total publications for each plant response variable studied in coastal dune systems globally. The WoS search tool was used to populate the list of studies used.

| Perfomance | N cases | N publications |
|---|---|---|
| Density | 40 | 10 |
| Growth | 39 | 10 |
| Survival | 42 | 8 |
| Richness | 17 | 4 |
| Reproductive output | 10 | 4 |
| Emergence | 4 | 2 |
| Occurrence | 8 | 2 |

**Table 2  Summary of meta-analyses results.** Summary of the mixed effect models used to examine the key factors on plant neighbor effects for plant density, growth, and survival in coastal dune systems.

| | Effect | DF | $R^2$(%) | Slope | P |
|---|---|---|---|---|---|
| **Density** | Geographic region | 2,37 | 0.7 | – | 0.33 |
| | MAP | 1,38 | 0.0 | −0.0003 | 0.67 |
| | NDVI—regional | 1,38 | 0.0 | −0.124 | 0.49 |
| | NDVI—local | 1,27 | 0.0 | 1.070 | 0.54 |
| | Neighbor life-form | 2,37 | 0.0 | – | 0.98 |
| | Target life-form | 3,36 | 0.0 | – | 0.88 |
| | Target life stage | 2,37 | 0.0 | – | 0.69 |
| **Growth** | Geographic region | 2,36 | 20.0 | – | **0.03** |
| | MAP | 1,37 | 0.0 | 0.001 | 0.13 |
| | NDVI—regional | 1,36 | 0.0 | −0.030 | 0.70 |
| | NDVI—local | 1,15 | 0.0 | −0.136 | 0.95 |
| | Neighbor life-form | 1,37 | 0.0 | – | 0.62 |
| | Target life-form | 1,37 | 0.0 | – | 0.40 |
| | Target life stage | 2,36 | 17.5 | – | 0.08 |
| **Survival** | Geographic region | 1,40 | 17.1 | – | **0.03** |
| | MAP | 1,40 | 0.0 | −0.001 | 0.40 |
| | NDVI—regional | 1,40 | 0.0 | −0.113 | 0.29 |
| | NDVI—local | 1,28 | 0.0 | −1.646 | 0.30 |
| | Neighbor life-form | 2,39 | 0.0 | – | 0.15 |
| | Target life-form | 2,39 | 0.0 | – | 0.36 |
| | Target life stage | 1,34 | 9.2 | – | 0.19 |

**Notes.**

DF, degrees of freedom; $R^2$, percentage of the true variation explained by the independent variable; Slope, only applicable for continuous predictors.

observed variability among study outcomes ($Q = 644$; df = 39, $P < 0.0001$). However, the variability in effect sizes for plant density were not explained by any of the factors we considered (Table 2). The presence of neighbors had a positive effect on the growth of the target species (Fig. 2A). The overall heterogeneity test was significant ($Q = 186$; df = 38, $P < 0.0001$) but only geographic region factor was a significant predictor of the variability amongst outcomes (Table 2). In the temperate and subarctic regions, there is no evidence of a neighboring effect on target growth (Fig. 2A). In the tropics, the presence of a neighbor increased the growth of target species (Fig. 2A). The survival of target plants was not significant affected by the presence of neighbors (Fig. 2B). The test of within-study heterogeneity was significant ($Q = 178$; df = 41, $P < 0.001$), but again only geographic region was a significant explanatory factor (Table 2). Target plant survival in tropical regions was increased by neighbours but not in the coastal dunes from temperate regions (Fig. 2B).

## DISCUSSION

Empirical studies support the hypothesis that facilitation between plants is an important ecological process in severe environments (*Brooker et al., 2008*) including as demonstrated

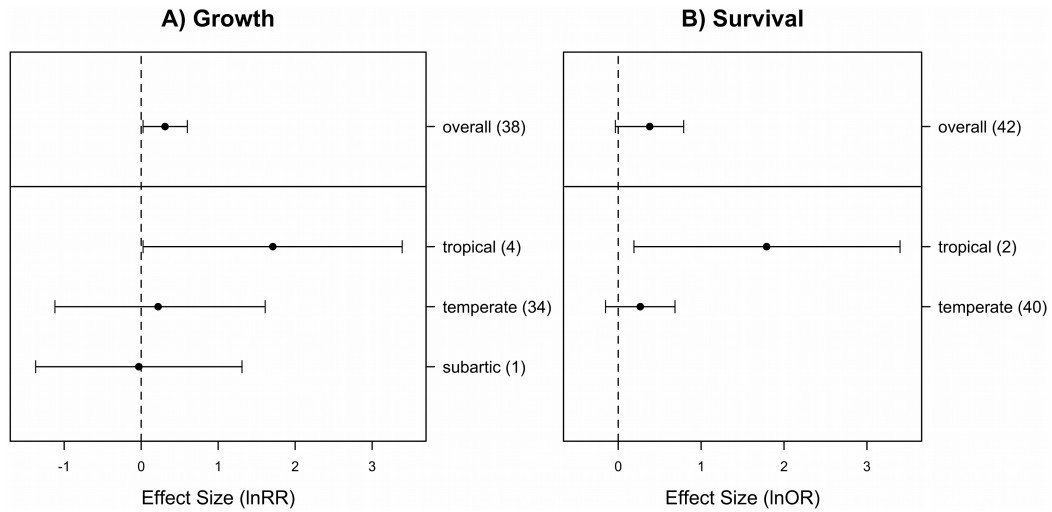

**Figure 2** **Mean effect sizes by geographical region, and the overall effect for (A) plant growth and (B) plant survival, in coastal dune systems.** Error bars are bias-corrected bootstrap 95% confidence intervals. The number of independent cases (outcomes) is shown in parentheses. Across all study cases, the presence of neighbor had a positive effect on overall growth but no effect on survival of the target species (given that confidence intervals that do overlap zero). Geographic region explained a portion of the variability among study cases (20% of target growth and 17% of survival). Although the small number of cases in the tropical region, neighboring plants increases the growth and survival of target plants in the tropics (confidence interval do not overlap zero), while no effect was observed in the temperate region and subartic (the last just in the case of growth).

herein coastal dunes. The first goal of the systematic review was to highlight general research gaps in the plant facilitation research in coastal dunes. We found a clear concentration of dune facilitation studies in temperate dunes, indicating that to assess the impact of climate differences and gradients on facilitation between plants, future studies need to be done in other dune systems such as the tropics. The systematic review also identified a predominance of observational studies over experimental studies, and this is unfortunate in many respects, given that the former is a weaker form of inductive inference. The second goal of this study was to examine the importance of factors quantitatively and contrast different scales of drivers on interaction strengths. The response variables measured in facilitation studies were an important determinant of the factors determining the strength of interactions. For the three response variables considered (density, growth, and survival), we found significant evidence for the importance of geographic region in determining the magnitude of facilitation, but no evidence for the effects of climatic and local factors within a region such as life form and life stage of the interacting species. Collectively, this indicates that facilitation is important in coastal dunes and that its relative intensity is best described by the regional context.

As highlighted previously, the clear concentration of dune facilitation studies in temperate dunes suggests that we need to expand the scope of coastal dune interaction studies to other geographic regions. This is important for a number of reasons. Macroecological
synthesis is an important, novel, and dominant source of theory validation in community ecology (*Keith et al., 2012*). Alpine and semi-arid syntheses (*Maestre, Valladares & Reynolds, 2005*; *Arredondo-Núñez, Badano & Bustamante, 2009*) and large-scale integrated experiments (*Fraser et al., 2013*) are a powerful means to test global issues including the importance of small-scale processes (*Paine, 2010*) such as interactions. Considering that coastal dunes are subject to significant global change effects (*Van der Meulen, Witter & Ritchie, 1991*), understanding how plant interactions vary between geographic regions increases predictive ecology on important issues such as climate change effects on plant community structure globally (*Michalet et al., 2014*). As highlighted by *Hesp (2004)*, even though few comparative studies have been carried out, differences in factors such as species, adaptative strategies and rates of plants growth indicate that ecological processes may be distinct between tropical and temperate dunes in many systems.

Another major limitation identified by the systematic review was the predominance of observational studies over experimental studies. Observational studies included spatial association analyses among species (*McIntire & Fajardo, 2009*; *Cushman, Waller & Hoak, 2010*; *Castanho, Oliveira & Prado, 2012*). Although positive associations provide evidence of facilitation, this associational pattern does not exclude alternative explanations such as shared physical microhabitats requirements and the tendency of some plants to act as foci for seed deposition (*Callaway, 1995*). Alternatively, experimental manipulations provide a causal form of verification for plant facilitation because the mechanistic pathways can be identified (*Callaway, 2007*). Consequently, we also recommend that coastal dunes be studied more comprehensively using manipulative approaches or a combination of observational and experimental methodologies to decouple direct from indirect effects (*Kunstler et al., 2006*), identify mechanisms (*Shumway, 2000*; *Maestre, Bautista & Cortina, 2003*; *Cushman, Lortie & Christian, 2011*), and examine the importance of local variation (*Lu et al., 2011*; *McIntire & Fajardo, 2014*). The extent that facilitation or plant interactions in general can be used to manage or restore highly impacted/stressed systems such as coastal dunes is generally best examined through manipulation.

The quantitative examination of plant facilitation magnitude across studies, i.e., the meta-analyses, showed that the factors influencing the occurrence and magnitude of facilitation in coastal dunes depended on the response variable measured. Whilst geographical region influenced the magnitude of facilitation for plant growth and survival, no effect of region was observed on interactions regarding plant density (Table 2). This result supports the general findings of another meta-analysis on facilitation for arid and semi-arid environments that concluded that the effect of abiotic stress on the outcome of interactions depended on the plant response (*Maestre, Valladares & Reynolds, 2005*). In order to explain this difference between response variables, we need to better understand how the neighbor presence changes the conditions and resources in its neighborhood and how it affects the distinct species-specific responses (*Michalet et al., 2014*). Therefore, to better understand the factors affecting the magnitude of positive interactions, we must investigate the mechanisms behind facilitative interactions in coastal dunes and, importantly, also record multiple target responses to neighbors (*Hastwell & Facelli, 2003*;

*Brooker et al., 2008*). This is rarely done in a single study (but see for instance *Rudgers & Maron, 2003*; *Cushman, Lortie & Christian, 2011*) but is nonetheless an important avenue of research that will benefit assessment of restoration efforts.

Growth and survival trends suggest that geographic region mediates the presence of facilitation more broadly in coastal dune systems. This is a very novel finding (*Thebault et al., 2014*). Altogether, these results showed that the presence of a neighbor was positive for plant survival and growth in the tropical region, whereas in the temperate and subarctic regions, the effects were neutral for both plant response variables. The environmental severity is relative to the stress tolerance and resource use adaptations of the species within a system (*Lortie, 2010*). The species composition and predominant life-forms differ between tropical and temperate dunes (*Hesp, 2004*). Therefore, the observed result can be the product of different sets of traits associated with the species in each region respectively, and consequently, distinct sensitivities to the changes in conditions and resources generated by neighbor presence in the tropical and temperate dunes. However, the limited number of tropical studies indicates that we need to be cautious in the generalization of this alternative hypothesis at this junction. In the present synthesis, we focused on only facilitation studies to test hypothesis related to the magnitude of this interaction, although competition and facilitation are of course both subsets of plant–plant interactions. To explore and contrast the relative importance of competition and facilitation, primary studies in coastal dunes must now test them directly and concurrently, and the scope of a synthesis must be expanded to also include plant competition studies.

The capacity for regional drivers of change to mediate positive, local interactions is a novel and important challenge to traditional community ecology and suggests that studies must also now consider the regional context in studying plant–plant interactions even at relatively fine scales in these systems. From a restoration and management perspective, this also suggests that best practices in using facilitation to reduce potential anthropogenic or disturbance effects may need to be tested and/or applied via different mechanistic pathways depending on the importance of regional drivers on productivity gradients and specific local limitations to key target plant species.

The use of remote sensing data together with meta-analytical techniques could be a powerful tool to explore the importance of climatic and environmental covariates (usually not provided by primary studies) driving ecological processes. In the present study we did not find an effect of NDVI and MAP on the variability of plant interaction in coastal dunes. One possible explanation for the failure to detect a significant effect of the remote sensing covariates and the other local covariates also tested (plant life-form and life-stage) is the highly variable nature of ecological data together with the small number of primary studies available to construct a big picture. However, as we accumulate primary data testing plant facilitation in coastal dunes, we should be able to drawn synthesis with more definitive conclusions about the factors driving facilitation intensity.

### Funding

We acknowledge the Canadian Bureau for International Education (CBIE) of Canada for the Emerging Leaders in the Americas (ELAP) Grant and São Paulo Research Foundation for the postdoctoral fellowship (FAPESP project No 2012/09794-7) provided to CT Castanho. This research was also funded by an NSERC Discovery Grant to CJ Lortie. The funders had no role in study design, data collection and analysis, decision to publish, or preparation of the manuscript.

### Grant Disclosures

The following grant information was disclosed by the authors:
Canadian Bureau for International Education.
São Paulo Research Foundation: 2012/09794-7.
NSERC Discovery Grant.

### Competing Interests

Christopher J. Lortie is an Academic Editor for PeerJ.

### Author Contributions

- Camila de Toledo Castanho conceived and designed the experiments, performed the experiments, analyzed the data, wrote the paper, prepared figures and/or tables, reviewed drafts of the paper.
- Christopher J. Lortie conceived and designed the experiments, wrote the paper, reviewed drafts of the paper.
- Benjamin Zaitchik contributed reagents/materials/analysis tools, reviewed drafts of the paper.
- Paulo Inácio Prado reviewed drafts of the paper.

### Supplemental Information

Supplemental information for this article can be found online at http://dx.doi.org/10.7717/peerj.768#supplemental-information.

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
