# Peer review of "A meta-analysis of plant facilitation in coastal dune systems: responses, regions, and research gaps"

_PeerJ, doi:10.7717/peerj.768_

## Round 0.1 · original submission · Major Revisions

· Academic Editor

Major Revisions

Three Reviewers have provided their comments about this manuscript.If you wish to revise your manuscript, please take the referee comments fully into account and provide point-by-point responses with a full list of changes.

Reviewer 1 ·

Basic reporting

L43-46: I don’t understand the context of this sentence. Please rephrase and clarify. If you would like to refer to the relative importance of the plant response versus the plant effects variable, you may consider Michalet et al. 2014 (FunctEcol) separating changes in plant-plant interactions along environmental gradients into environmental severity effects (i.e. plant response variable) and benefactor trait effects (i.e. plant effects variable).
L133: Delete one “to”
Table S1 and S2: It looks like there is a study ‘Oriol et al. 2010’ included in the meta-analysis, but not in the systematic review. Does that make sense? Or is Oriol et al. 2010 the same study as Grau et al. 2010?
L182: Please consider providing the critical thresholds of the fail safe numbers for each dataset in order to justify which dataset shows limited and no bias respectively.
Fig. 2A: I wonder if a non-linear relationship would be more appropriate here. Is there a possibility to test for non-linear relationships?
Fig. 2-4: Do these graphs show partial effects of the individual predictors, i.e. after accounting for the effects of the other predictor variables included in the model? I think showing partial effects would be most appropriate here.
L245ff: It is really hard for me to imagine an influence of NDVI on the magnitude of facilitation. I think you mean that NDVI represents a driver that may also influence facilitation. This needs to be clarified and discussed in more detail.
L257-258: Do you use NDVI as a measure of vegetation biomass? This should be clarified in the methods and supported by data or references.

Experimental design

I miss a link in the methods section justifying the use of the selected explanatory variables (in particular mean annual precipitation and NDVI), given that the main stressors and disturbances affecting plant facilitation in costal dune systems are “soil moisture, soil nutrient, wind exposure, sand burial, salt spray and soil salinity” (L33). This should be clarified.
Overall, my main critique would be the selection of explanatory variables, in particular the selection of the regional explanatory variables. They do not seem to be nicely linked to the main stressors for coastal dune systems, such as soil moisture, soil nutrients, sand burial and salinity. This may also explain the overall weak relationships and the low variance explained. I think this deserves some revision (if such data would be available) or at least discussion.

Validity of the findings

No Comments

Reviewer 2 ·

Basic reporting

Ln 61-62, “queried using ISI Web of Science … by using …”. This is a bit confusing

Ln 73-74, How data were selected when the study has a second treatment factor (like water addition).

Ln 80, Target species is always a single species or can be a bunch of several species? If the author removed a nurse shrub and reported the biomass of each individual species separately, how such kind of data were analyzed in your study?

Ln 82, what “young” here exactly mean?

Ln 83, for experimental studies, was regrowth of facilitators always removed repeatedly?

Ln 120, “categorical responses” should be binomial responses?

Ln 129, “once it included at least 5 independent studies”, awkward

Ln 444, “A summary”, remove “A”

Experimental design

No Comments

Validity of the findings

This paper presents some interesting findings. I like the way the author conduct the research and analysis. I have a couple of major concerns.

1) The authors used actually very few data to examine large-scale patterns. For either density, growth or survival, they had only about 40 effect sizes from just about 10 studies. This is actually a very small number for any meta-analysis that aims to address macro ecological patterns (many other factors are actually highly variable among studies, e.g. plot size, species, sand dune zone). The relationships between facilitation and MAP/NDVI found in the manuscript were quite variable. Even when they found a significantly positive or negative relationship, the variance explained by the mediator was very low (< 0.06). I really don’t think those relationships make much of sense. Should be cautious to interpret those results.

The authors also claimed that they found geographical region to be an important factor affecting facilitation. But with just 2-4 effect sizes from tropics, what’s the power of the meta-analysis? Yes, the authors recognized this limitation in Discussion. But this limitation should be further addressed like in the legend of the figure. In a number of cases readers do not go through the paper and then cite it as evidence, so important limitations should be made as clear as possible.

2) This meta-analysis was called a meta-analysis of “facilitation”. But having seen their data, I found nearly half of their effects sizes show strong competition. Their overall effect sizes were also generally insignificant. It’s entirely OK to present insignificant results. But the authors did not interpret what those insignificant results meant for sand dune community organization? Does this mean that facilitation is unimportant or highly variable in sand dunes?

The authors should recognize that species interactions always change between facilitation and competition, and that competition has been found in many facilitation studies. Similarly, facilitation can be found in a number of competition studies, too. So it is not really meaningful to separate competition and facilitation in a meta-analysis, particularly when the responsible variable was target plant performance. This means that the authors’ way for literature search can be problematic, because they only searched for“facilitation” or "positive interaction*, but did not search for competition studies.

3) Zone of sand dunes. Perhaps, a very important factor determining facilitation in sand dune communities is distance from shore. Experiments or observations can be conducted in different zones varying from fore zones to back zones. Are plant interactions examined in fore zones more likely to be positive, while those examined in back zones negative? Plant facilitation papers can examine interactions in several different zones. It’s unclear how the authors treated this factor in their data selection and analysis.

Comments for the author

Your paper is a nice one. I really like your general methodology. Sand dune is a great study system, too. But you did not really have enough data to make conclusive statements on the mediators or macro ecological patterns of facilitation. I encourage the authors further address any limitations to avoid misleading use/citations of the weak, uncertain evidence presented in your paper post publication.

Reviewer 3 ·

Basic reporting

The article appears to follow journal guidelines, and basic reporting seems fine.

Experimental design

The article is a meta-analysis rather than an original experiment, but the sampling and analysis of existing data sets is clearly described.

Validity of the findings

Overall the findings appear valid, but there were some issues with writing and organization of the Results and Discussion sections that made this difficult to evaluate. These issues are described below under General Comments.

Comments for the author

The manuscript describes a meta-analysis of facilitation in coastal dune plants. There have been previous reviews of facilitation in general and of facilitation in other systems, but no previous reviews and meta-analyses of facilitation in coastal dunes, making this a novel contribution to the field.
Although the authors are justified in limiting the scope of the study to work on facilitation, the authors should at least discuss competition. For example, is there any evidence for competition among coastal dune plants? Is facilitation or competition stronger in dunes? Could there be bias caused by researchers intending to study competition but not finding or reporting it because they are actually finding facilitation, or facilitation balanced by competition (leading to non-significant net interactions)? Because competition and facilitation lie along a continuum of interaction strength and direction, focusing on only one end of the spectrum can give a limited view of plant interactions. This should be discussed.
The Results section was too brief and difficult to follow. I recommend the following changes to make this more clear. At the end of the Introduction, give the main hypotheses or aims of the study. In the Results section, provide, in the same order, the main findings related to these hypotheses or aims. The Results should be given in a logical order. They should also be written more clearly so that it is easier for the reader to determine what the main results are. For example, the first sentence of the third paragraph of the results states ‘The overall effect of the neighbor presence was neutral for plant density response (i.e.95% CI crossed zero), and the overall heterogeneity test was significant indicating that the different cases do not share a common effect (QT=106.3; df= 39, P=0.00000).’ I’m not sure what this means. Possibly the authors intend to say something like ‘There was no evidence for facilitation based on target plant density, given the fact that density response was not significantly different from zero based on 95% confidence intervals. Furthermore, a test of heterogeneity showed that different cases involving density response did not share a common effect, indicating that facilitation does not appear to drive density response in target plants.’ Although this might not be exactly what the authors are trying to say (since this was difficult to determine), the point is that the authors should be much more clear. The Results section should also be expanded, with more detail given. The writing in the results section needs to be greatly revised and improved before the manuscript can even be evaluated.
After the Results section is revised, the Discussion also needs to be revised to discuss the main results in a logical and coherent order. Currently the Discussion alternates between discussing characteristics of the studies (such as the proportion observational compared to experimental, and the proportion in temperate compared to tropical systems), and discussing the main findings of the analyses. Also, the Discussion currently makes claims without adequate support or evaluation. For example, the Discussion states that region and climate were more important than life form or life stage in determining the magnitude of facilitation. I don’t think it’s possible to draw this conclusion based on the fact that some significant effects are seen for some of the regional variables but not for stage or life form (Table 2). I think it’s safer to say that there is evidence for the importance of regional factors in determining facilitation, but no evidence for effects of life form or stage (rather than regional factors being more important than life form or stage). The authors make strong claims in the abstract and manuscript about regional factors being more important, but I don’t think this is fully supported by the data, and they are overstating their case.
The Introduction could also be reorganized and revised. After lines 17-19 listing other systems for facilitation research, then include the information from lines 31-34 about facilitation research conducted on coastal dunes. Drop the sentence in lines 19-21, which makes it sound as if research on facilitation has taken place in other systems but not coastal dunes, when the whole point of the paper is to review dune facilitation research. The Introduction should be organized by major questions about facilitation addressed in this analysis, such as the degree to which facilitation occurs on coastal dunes, the importance of local versus regional processes, and other factors such as life stage. These questions should frame the introduction and lead to a set of specific hypotheses addressed by this research, which are listed at the end of the introduction.
I don’t think Figure 1 is really needed or adds more to the information already provided in the text. For the legends of the other figures, it would be helpful if these gave more information about the main point or finding of the result shown.
In the Abstract, change ‘Consequently, plant facilitation studies in coastal dunes are dependent on the response variable measured...’ to ‘Consequently, conclusions about plant facilitation in coastal dunes depend on...’.

---

## Round 0.2 · Major Revisions

· Academic Editor

Major Revisions

Two original reviewers agree that your revisions substantially improved the manuscript. The third one declined to review the revised manuscript. While the responses to the prior reviewers seemed sufficient I believe that there was still issues in the Experimental design section, so I invited a third reviewer to evaluate your work.

The major problems noted in first round of reviews have been resolved, but there are a few remaining issues regarding the data search that need to be addressed before this manuscript is ready for publication. Please address the reviewer comments in your revised manuscript and accompanying cover letter. I also invite you to carefully proof read and edit your manuscript before your final submission.

Reviewer 2 ·

Basic reporting

The manuscript still contain a number of grammatical or spelling errors. Here are some of them:

ln 35, 278, environmental severity?

ln 41, coastal dunes, not costal dunes

ln 81, wich should be which?

ln 88, 98, 99, "a indicator"?

ln 170, arctic-subarctic, not artic-subartic

ln 191, Please clarify "The fail-safe numbers indicated limited to no bias"

ln 199, none of the explanatory variables we examined?

ln 207, "had no effect.(Fig. 2B)."

ln 209-210, Awkward sentence.

ln 229-231, you already said so in the above paragraph.

ln 289, primary studies of coastal dunes?

Experimental design

ln 61, Why not include a sentence here saying that "we did not consider plant competition studies in sand dunes, because ..."

Validity of the findings

The manuscript has limitations. The authors have clarified some of these limitations in their revision.

Comments for the author

The manuscript is improved.

Reviewer 3 ·

Basic reporting

See previous review.

Experimental design

See previous review.

Validity of the findings

See previous review.

Comments for the author

It appears that the authors have adequately addressed reviewer concerns and suggestions. Although some limitation of the study remain, these are sufficiently disclosed and discussed. I think this manuscript makes an important contribution to our understanding of facilitation in coastal dune plants, and provides a novel overview and analysis of studies on this topic.

Reviewer 4 ·

Basic reporting

I feel that the manuscript would benefit from careful editing by a native English speaker (see e.g. lines 47-53).

Experimental design

The most serious methodological problem with this study is related to the data search. There is a continuum of plant-plant interactions, from strongly negative (competition) to strongly positive (facilitation). Using only keywords “facilitation” and “positive interaction*”, the authors disregarded all studies that demonstrated competition, or found neutral relationships between plants. This happened when the authors of primary studies tested for competition effect and did not mention facilitation in the title and/or abstract of their study. Consequently, the outcomes of this meta-analysis are biased towards facilitation. This is an obvious example of research bias, when the authors pre-define the outcomes of their study at the data collection stage. Keeping in mind relatively low number of studies used in meta-analysis, I strongly suggest that the authors expand the scope of their research and conduct additional data search using “competition”, “negative interaction*” and “plant-plant interaction*”.

Furthermore, the ‘technical’ quality of the meta-analysis presented in the submitted manuscript is relatively low. The manuscript meets only a fraction of quality criteria required for modern ecological meta-analyses (Koricheva & Gurevitch, J.Ecol. 2014, 102: 828-844). In particular, although more than one estimate of effect size per study was included in the analysis, potential non-independence of these estimates had not been taken into account. Similarly, potential non-independence of and interactions between moderators have not been accounted. The phylogenetic relatedness of species was not considered. No sensitivity analysis was performed to assess the impacts of methodology (e.g., observational vs experimental study, long-term vs short-term study, plot size, species richness in the community etc.) on the outcomes of primary studies.

I am unhappy with separate meta-analyses performed for individual response variables (eight in total). To me, it is more appropriate to combine all data sets and conduct an overall analysis, asking in particular whether different response variables show uniform effects; now the conclusion that “the occurrence and magnitude of facilitation in coastal dunes depended on the response variable measured” (lines 261-262) is not confirmed statistically. Only if the responses differ among variables, or among the groups of variables, the separate analyses are justified (assuming that non-independence of different response variables calculated from the same study is accounted for).

It is not clear how the authors analyzed the impacts of continuous variables on the effect sizes, because the use of meta-regression is not mentioned.

I would suggest to clearly define the study system, coastal dunes, and to include some criteria on habitat selection in the manuscript. Now it seems that the authors simply followed the terminology used in primary studies, which may well be inconsistent. There are several classifications of dunes (e.g., parallel dunes, blowouts, parabolic and transgressive dunes) – do all types of dunes show the same patterns in plant-plant interactions? Were all data collected at sea shores, or there were some data from lake and river shores involved? Some information on distribution of coastal dunes across the geographical regions would also be helpful (in connection with the detected predominance of studies conducted in temperate regions).

There are many more sources of variation that can explain the differences among studies (e.g., native vs introduced species, slow vs fast growing plants, mycorrhizal vs non-mycorrhizal plants, etc.). The authors may consult several recent meta-analyses addressing different aspects of plant ecology to expand their list of classificatory variables.

I am confused by a contradiction between the numbers of effect sizes reported in lines 172 (362 ES) and 184 (160 ES). Does the first value refer to verbal conclusions made in original publications? If yes, then it is more appropriate to use a term ‘votes’ (or ‘verbal conclusions’) instead of effect sizes in line 172. However, analyses based on vote counting have repeatedly been criticized. More generally, I doubt that a “Systematic review” part adds any value to the present manuscript, because 20 of 32 studies used in this part are included in meta-analysis. From my point of view, it would be more meaningful to compare between the studies included in meta-analysis (20) and not included in it (12), rather than to provide pooled information on these 32 studies.

Validity of the findings

Keeping in mind the criticism expressed above, I doubt that the main conclusion of this study, namely “that facilitation is important in coastal dunes and that its strength is… shaped by regional context” is substantiated. Inclusion of “negative” (in terms of facilitation hypothesis) studies is needed to rigorously test this hypothesis.

Comments for the author

In studies of plant-plant interactions in a certain environment, the relative importances of competition and facilitation should always be taken into account, and the hypothesis that ‘facilitation is important…’ should be rephrased accordingly (when terms ‘competition’ and ‘plant-plant interactions’ are included to search of original studies.

---

## Round 0.3 · accepted · Accept

· Academic Editor

Accept

Thank you for your revised manuscript "A meta-analysis of plant facilitation in coastal dune systems: responses, regions, and research gaps." I am pleased to tell you that your work has now been accepted for publication in PeerJ.